# Phenolic Acids and Flavonoids from *Pithecellobium dulce* (Robx.) Benth Leaves Exhibit Ovicidal Activity against *Haemonchus contortus*

**DOI:** 10.3390/plants11192555

**Published:** 2022-09-28

**Authors:** Agustín Olmedo-Juárez, Ana Laura Jimenez-Chino, Alejandro Bugarin, Alejandro Zamilpa, Pedro Mendoza-de Gives, Abel Villa-Mancera, María Eugenia López-Arellano, Jaime Olivares-Pérez, Edgar Jesús Delgado-Núñez, Manases González-Cortazar

**Affiliations:** 1National Centre for Disciplinary Research in Animal Health and Innocuity (CENID-SAI), National Institute for Research in Forestry, Agriculture and Livestock, INIFAP, Carr. Fed. Cuernavaca-Cuautla No. 8534, Jiutepec CP 62550, Morelos, Mexico; 2Biotechnology Engineering, Polytechnic University of the State of Morelos, Boulevard Cuauhnáhuac #566, Col. Lomas del Texcal, Jiutepec CP 62550, Morelos, Mexico; 3Department of Chemistry and Physics, Florida Gulf Coast University, Fort Myers, FL 33965, USA; 4South Biomedical Research Center, Social Security Mexican Institute (CIBIS-IMSS), Argentina No. 1, Xochitepec CP 62790, Morelos, Mexico; 5Faculty of Veterinary Medicine and Zootechnics, Meritorious Autonomous University of Puebla, 4 Sur 304 Col. Centro, Tecamachalco CP 75482, Puebla, Mexico; 6Faculty of Veterinary Medicine and Zootechnics, Autonomous University of Guerrero, Altamirano, Km 3.0 Altamirano-Iguala Highway, Colonia Querendita, Altamirano CP 40660, Guerrero, Mexico; 7Faculty of Agricultural, Livestock and Environmental Sciences, Autonomous University of Guerrero, Iguala CP 40040, Guerrero, Mexico

**Keywords:** *Pithecellobium*, pinzan, legume, flavonoids, anthelmintic activity, High Pressure Liquid Chromatography (HPLC)

## Abstract

*Pithecellobium dulce* (Robx.) Benth is an arboreal legume used in traditional medicine for the treatment of several ailments, including a number of intestinal disorders, and as a natural deworming. The objective of this study was to evaluate the ovicidal activity of a hydroalcoholic extract (HA-E) and its fractions (aqueous, Aq-F and organic, EtOAc-F) from *P. dulce* leaves, as well as subfractions (C1F1–C17) obtained from EtOAc-F against *Haemonchus contortus* eggs. The HA-E, Aq-F, and EtOAc-F were assessed at 0.03–5.00 mg/mL and subfractions (C1F1–C17) were assessed at 0.06–1.00 mg/mL. The HA-E and Aq-F showed an ovicidal activity close to 100% at 2.5 mg/mL, and EtOAc-F displayed the highest anthelmintic effect (100% at 0.25 mg/mL). Meanwhile, the sub-fractions with the highest ovicidal effect were C1F6, C1F9, C1F10, C1F11, and CIF13. The main compounds identified in the most active fractions (C1F9, C1F11, and C1F13) were kaempferol (**1**), quercetin (**2**), coumaric acid (**3**), ferulic acid (**4**), luteolin 7-O-rhamnoside (**5**), quercetin 3-O-rhamnoside (**6**), and a caffeoyl derivate (NI). The results indicate that *P. dulce* leaves exhibit a potent anthelmintic activity and contain bioactive compounds able to inhibit egg hatching in *H. contortus*. Therefore, this plant could be used for the control of gastrointestinal nematodes in small ruminants.

## 1. Introduction

Small ruminants feeding under grazing conditions are exposed to several diseases caused by different pathogenic agents, i.e., viruses, bacteria, and parasites that affect the livestock industry [1]. Gastrointestinal nematodes (GIN) are important parasites with high prevalence in grazing animals in primarily tropical and subtropical regions. *Haemonchus contortus* is one of the most prominent pathogenic parasitic nematodes affecting sheep and goats. Due to its hematophagous habits, *H. contortus* can cause severe damage to the animals, thus affecting their zootechnical potential [2]. In livestock, infections with *H. contortus* are difficult to control, resulting in high economic losses worldwide (Corwin, 1997; Rodríguez-Vivas et al., 2017) [3,4]. In addition, the frequent and continuous use of chemical anthelmintic drugs has led to anthelmintic resistance worldwide. Recently, there has been a growing number of reports of resistance to commercially available antiparasitic drugs [5,6]. In this context, both the search for alternatives to control GIN and studies about plants/plant extracts rich in secondary metabolites with potential anthelmintic effect have shown encouraging results [7]. Some plants of the family Fabaceae have been assessed against GIN, including *H. contortus*, with promising results [8,9,10]. In Mexico, some arboreal legumes, such as *Lysiloma acapulcensis*, *L. latisilicum*, *Leucaena leucocephala*, *Caesalpinia coriaria*, *Acacia cochliacantha*, *A. farnesiana*, and *Prosopis laevigata*, are being explored for their potential anthelmintic activity, also with promising results. These plants contain phenolic compounds with nematocidal activity, such as condensed tannins, myricitrin, gallic acid, coumaric acid, isorhamnetin, caffeoyl and coumaroyl derivatives, and galloyl flavonoids [8,11,12,13,14]. *Pithecellobium dulce* (Robx.) Benth is an arboreal legume commonly named in Mexico as “guamuchil” or “pinzan” that is used in traditional medicine as a remedy against indigestion, to treat bilious disorders, and as an anthelmintic (e.g., to cure malaria) [15]. There are reports of its antioxidant, antimicrobial, and anthelmintic activities [16,17,18]. Aqueous and methanolic extracts of *P. dulce* leaves have been reported to have anthelmintic effect against GIN in small ruminants [19,20]. Thus, this study assessed the ovicidal activity of a hydroalcoholic extract (HA-E), two fractions (an aqueous fraction, Aq-F, and an organic fraction, EtOAc-F), and sub-fractions from *P. dulce* leaves against *H. contortus* eggs under in vitro conditions.

## 2. Results

### 2.1. Egg Hatching Inhibition (EHI) Test

The ovicidal activity of the HA extract and the Aq and EtOAc fractions, as well as their proper controls, are shown in Table 1. The egg hatching inhibition percentage of the HA-E and the Aq fraction were greater than 90% from 0.62 mg/mL concentration. The organic fraction (EtOAc-F) exhibited the best ovicidal effect (100%) at 0.25 mg/mL. The negative controls showed no egg hatching inhibitory effect, and thiabendazole displayed a total ovicidal activity.

Table 2 shows the effective concentrations (50 and 90) required to cause an ovicidal effect of the HA extract and both Aq and EtOAc fractions. The latter fraction had the best anthelmintic effect, with EC_50_ and EC_90_ values of 0.08 and 0.14 mg/mL, respectively.

The EHI percentages of subfractions C1F6-C1F13 are shown in Table 3. The highest ovicidal effect was observed in the subfractions C1F9, C1F10, C1F11, and C1F13, with 100% of EHI at 1 mg/mL.

Table 4 shows the effective concentrations (50 and 90) of subfractions with a concentration-dependent effect. The C1F9 and C1F11 fractions were the most active ones, with EC_50_ = 0.03 and 0.04 mg/mL, a EC_90_ = 0.06 and 0.07 mg/mL, respectively.

### 2.2. Identification of Compounds

The HPLC chromatograms of the HA-extract as well as the Aq and EtOAc fractions are shown in Figure 1. The major compounds identified were coumaric acid with a retention time (rt) of 10.01 min and a UV absorption spectrum at λ_max_ of 227.4 and 310.0 nm; and luteolin 7-O-rhamnoside (**5**) with a rt = 11.07 min and a UV absorption spectrum of 209, 263.9, and 343.4 nm. The percentages of the area under the curve of coumaric acid and luteolin 7-O-rhamnoside in the hydroalcoholic extract were 23.56 and 76.44%, respectively. In the aqueous fraction they were 38.25 and 61.75%, while for the ethyl acetate fraction were 19.77 and 80.23%, respectively.

Figure 2 shows the chemical constituents present in the bioactive subfractions. The major compounds identified in C1F9 were kaempferol (**1**) with a retention time (rt) of 21.8 min (75.87%) and a UV absorption spectrum at λ_max_ of 209.9, 265, and 363 nm; a compound derived from caffeic acid (unidentified) of 12.5 min (2.29%) and a UV absorption spectrum at λ_max_ of 219.2, 242.7, and 325.5 nm; and coumaric acid (**3**, rt= 10.01, 21.84%)**.** The C1F10 subfraction showed the presence of (**1**, rt = 21.94, 25.96%), ferulic acid (**4**, rt = 10.4 min, 4.39%) and UV absorption spectrum λ_max_ of 218.1, 234.5, and 320.8 nm and (**3**, rt = 10.01 min, 69.65%). In subfraction C1F11, the presence of kaempferol (**1**, rt = 21.94, 8.47%), quercetin (**2**, rt = 14.96 min, 8.79%), (**4**, rt = 10.4 min, 19.75%), and (**3**, rt = 10.1 min, 62.99%) were observed. The subfraction C1F13 contained luteolin 7-O-rhamnoside (**5**, rt = 11.1 min, 90.02%) and coumaric acid (**3**, rt = 10.1 min, 9.98%), while subfraction C1F15 contained luteolin 7-O-rhamnoside (**5**, rt = 11.1 min, 93.05%) and quercetin 3-O-rhamnoside (**6**, rt = 10.03 min, 6.95%). Fingerprint comparison of the bioactive fractions with commercial references (kaempferol, quercetin, coumaric acid, ferulic acid, and quercetin 3-O-rhamnoside) allowed identification of these polyphenols (Appendix A). 

Compound (**1**): analysis of the acquired NMR spectra [^1^H, ^13^C, and DEPT (Appendix A)] and its comparison to data reported in the literature [21], this compound is a flavonoid known as Kaempferol (**1**). Compound **1** displayed a quasimolecular positive ion at *m/z* 287 [M + H]^+^ in ESI-MS (Appendix A), with molecular formula C_15_H_10_O_6_.

Compound (**5**): this compound exhibited a quasimolecular negative ion at *m/z* 432 [M]^−^ in EI-MS (see Appendix A), with the molecular formula C_21_H_20_O_10_. According to electrospray ionization mass spectrum (ESI-MS) analysis, this compound corresponds to a glycosylated flavonoid called luteolin 7-O-rhamnoside.

The chemical structures of the isolated compounds are shown in Figure 3.

## 3. Discussion

The nematode *H. contortus* is one of the most pathogenic agents that severely affects the health and production of small ruminant herds. In this scenery, farmers deworm their animals with anthelmintic drugs in a constant and indiscriminate manner, leading to the imminent presence of anthelmintic resistance by the parasites [22]. The use of medicinal plants represents a viable way to discover new molecules for controlling GIN in ruminant herds. The results found in the present study revealed that *P. dulce* leaves exhibited the highest ovicidal effect against the parasite *H. contortus*. The HA extract displayed a concentration-dependent effect, with an ovicidal effect close to 100% at 0.62 mg/mL. This finding demonstrates that the anthelmintic potential of this legume and the ovicidal effect of the HA extract were higher than those of other plants belonging to the same family. For instance, Zarza-Albarrán et al. have shown an ovicidal effect close to 100% for a hydroalcoholic extract from *Acacia farnesiana* pods against *H. contortus*, using a concentration of 25 mg/mL [14]. In another study, a HA extract obtained from *Caesalpinia coriaria* fruits was tested against a miscellanea of cattle gastrointestinal nematodes achieved 97% of EHI at 1.56 mg/mL [23]. In the present study, the effective concentration of the HA-E (EC_50_ = 0.18 mg/mL) was 72.05 times more active than the one shown by *A. farnesiana* (EC_50_ = 12.97 mg/mL) pods HA-E and 5.11 times more effective than *C. coriaria* HA-E fruits (EC_50_ = 5.11 mg/mL).

After the corroboration of the ovicidal activity of the *P. dulce* full extract and the organic and the aqueous fractions by bipartition were obtained, a high nematocidal effect was observed for EtOAc-F. After analysis, the EC_50_ of the EtOAc fraction and comparing it with the HA-E and the aqueous fraction (Aq-F), it was noted that the organic fraction was 12.15 times more active than the other fractions (HA-E and Aq-F). The same analysis was performed with the EC_90_, showing that the EtOAc-F was 23.85 times more active than the HA extract and the Aq fraction. Similar results have been reported for other arboreal legumes such as *Lysiloma acapulcensis* and *A. cochliacantha*, whose organic fractions (EtOAc-F) were assessed against *H. contortus* eggs, resulting in 94.85% activity at 6.2 mg/mL and 98% at 1.56 mg/mL [12,13]. The liquid-liquid extraction of the HA extract with ethyl acetate, performed in this study and in other studies with legumes, provides the basis for the determination of the biological activity of hydroalcoholic extracts of plant materials from several arboreal legumes. Thus, the constituents present in the EtOAc fraction from *P. dulce* leaves and their ovicidal activity corroborated this asseveration.

The chromatographic process of EtOAc-F fractionation permitted the isolation of eight bioactive subfractions with an important ovicidal effect. This finding could be explained by the presence of known compounds associated with nematocidal activity, such as kaempferol, ferulic acid, coumaric acid, kaempferol rhamnoside, and quercetin. Similarly, Castillo-Mitre et al. tested a mixture of *p*-coumaric acid and ferulic acid obtained from *Acacia cochliacantha* leaves against *H. contortus* eggs, and demonstrated an ovicidal activity close to 100% at 1 mg/mL [12]. These compounds are present in the C1F11 (Figure 2), and this fraction exhibited a total ovicidal activity at 1 mg/mL (Table 3).

Plants belonging to the Fabaceae family are rich in secondary metabolites such as tannins, flavonoids, and other phenolic compounds [24]. Several studies with arboreal legumes and their secondary compounds have demonstrated an important nematocidal activity against gastrointestinal parasitic nematodes of cattle and small ruminants. This represents a potential strategy for controlling parasitosis, providing a different perspective to the use of anthelmintic drugs chemically synthesised in a laboratory. Several studies have demonstrated that some phenolic compounds play an important role against different stages of GIN. The phenolic compounds produced by other arboreal legumes, such as coumaroyl and caffeic derivatives, caffeic acid, gallic acid, isokaempferide, 4-5-di-*O*-caffeoylquinic, and naringenin 7-O-(6-galloylglucoside), have contributed to the finding of novel drugs obtained from natural sources, facilitating the development of new anthelmintics for controlling GIN of importance in the livestock industry [14,25].

The arboreal legume *P. dulce* has been used in traditional medicine to treat gastrointestinal disorders and malaria [15,26,27]. In a study carried out by Olmedo-Juárez et al., a crude extract complex with fresh leaves from *P. dulce* was assessed against a GIN strain, and the authors achieved an ovicidal effect close to 40% at 0.5 mg/mL [19].

The clear evidence of nematocidal activity found in the present study with *P. dulce* leaves allowed the identification of several bioactive compounds, such as ferulic acid, coumaric acid, kaempferol rhamnoside, and quercetin (Figure 2 and Figure 3).

Analysing the HPLC sub-fractions showed that the major compounds in one more of the active treatments (C1F11) correspond to ferulic and coumaric acids; meanwhile, quercetin and kaempferol were the compounds found in minor proportions (Figure 2 and Figure 3). The mixture of these compounds could exert a synergic ovicidal effect. On the other hand, analysing the compounds present in the C1F9 subfraction led to the inference that kaempferol was the major constituent and ferulic acid the minor. However, when kaempferol was tested individually, it showed no ovicidal activity against *H. contortus* eggs (Table 2). Nevertheless, the mixture of ferulic acid and kaempferol, as present in C1F9, displayed the highest anthelmintic effect (Table 3).

Several studies with pure molecules, obtained commercially or directly obtained from plants, indicate two anthelmintic effects (synergism or additive). For instance, Mancilla-Montelongo et al. assessed seven hydroxycinnamic acid derivatives, either individually or combined, against *H. contortus* eggs and infective larvae [28]. The authors found that ferulic and chlorogenic acids blocked egg hatching at 0.24 and 0.52 mg/mL, respectively. In the same study, when testing ferulic acid and coumaric acid individually, the authors only found ovicidal activity for ferulic acid. Likewise, these authors performed a combination of ferulic, chlorogenic, cinnamic, coumaric, and caffeic acids and reported a high anthelmintic effect (EC_50_ = 0.923 mg/mL). In another study, phenolic compounds quercetin, rutin, coumarin, and caffeic acid were assayed against eggs of the cattle nematode *Cooperia punctata* [27]. A synergic effect was observed with the combination of caffeic acid with coumaric acid, quercetin, and rutin [29]. These studies, combined with the findings of the present work, support the hypothesis that the anthelmintic effect is attributed to the hydrocinnamic acid derivatives. Nonetheless, future studies should be performed using the bioactive subfractions from *P. dulce* leaves to determine the amounts of ferulic and coumaric acids and kaempferol, along with their relations to the ovicidal activity on *H. contortus* eggs.

According to reports related to anthelmintic activities of several phenolic compounds through in vitro and in vivo studies, and considering the results found herein, the importance of medicinal plants as potential natural anthelmintics is evident. It should be note that the use of extracts or fractions depends on multiple factors such as environmental conditions, phenological stage and plant age, among others. Likewise, the extraction method of the secondary metabolites is associated with the affinity of the solvent used. Thus, the standardisation of these extracts or fractions, according to the contents of bioactive compounds, is necessary to obtain reliable results for controlling *H. contortus* and other GIN affecting herds worldwide. The in vitro efficacy of the above-mentioned molecules should be tested under in vivo conditions in animal farms to corroborate their anthelmintic effects, thereby improving animal health and increasing farm productivity.

## 4. Materials and Methods

### 4.1. General

The chemical compounds used in this study were analytical grade. The solvents (ethanol, methanol, ethyl acetate and distilled water), dimethyl sulfoxide (DMSO), *p*-coumaric acid (≥98%), ferulic acid (≥99%), quercetin (≥95%), and thiabendazole (≥98.0%) were purchased from Merck KGaA^®^ (Darmstadt, Germany) and Sigma-Aldrich (St. Louis, MO, USA). Reagents for the bioassays (ovicidal activity) were purchased from Corning^®^ (Corning, NY, USA). HPLC analyses were performed on a Waters 2695 Separation module system, equipped with a Waters 996 photodiode array detector and the Empower Pro software (Waters Corporation, Milford, MA, USA). Mass spectroscopy was performed on a Waters Xevo TQD mass spectrometer with an ESI ion source (Waters Milford, Milford, MA, USA). The ultraviolet (UV) spectra were obtained using a Waters array detector (Waters Co. 2996, Milford, MA, USA). Thin-layer chromatography (TLC) was performed using TLC Silica gel 60, F254 and 20 × 20-cm aluminium sheets (Merck KGaA^®^, Darmstadt, Germany).

### 4.2. Plant Material

*Pithecellobium dulce* leaves were collected from the San Gaspar municipality, Jiutepec, Morelos, Mexico (18°52′53′′ N and 99°10′40′′ W). Fresh leaves (young and mature) were collected between September and October 2018. One specimen was deposited in the herbarium of the Centre for Research in Biodiversity and Conservation at Universidad Autónoma del Estado de Morelos, Mexico, under voucher code 33948. The name of this plant has been checked in the data bases of the plant list [30]. The plant material was dried at room temperature in the dark for 3 weeks, and subsequently, the leaves were ground using an electrical mill (Wiley mill, TS3375E15) to reduce the size particle to 4–6 mm.

### 4.3. Hydroalcoholic Extract

Dried leaves (300 g) were subjected to liquid extraction using distilled water (2100 mL) and methanol (900 mL) at room temperature for 24 h, and the liquid extract was filtered using different sieves (gauze, cotton, and Whatman^®^4 filter paper). The hydroalcoholic extract (HA-E) was concentrated under reduced pressure using a rotary evaporator (55 °C, Büchi R-300, Flawil, Switzerland) to obtain a semisolid extract, which was finally freeze-dried, resulting in a brown powder (54 g, 18%).

### 4.4. Fractionation of the Hydroalcoholic Extract (HA-E)

The HA-E (50 g) was suspended in distilled water (2500 mL) and EtOAc (2500 mL) in a separatory funnel, resulting in two fractions: an aqueous (Aq-F) and an organic fraction (EtOAct-F). The solvents of both fractions were removed using a rotary evaporator and finally dried via lyophilisation, producing residues of 6.18 and 3.1 g, respectively. The HA-E and fractions were stored at −40 °C until pharmacological and phytochemical analysis could be performed [25].

### 4.5. Fractionation of the Organic Fraction (EtOAc-F) and Identification of Compounds *(**1**–**6**)*

The organic fraction (9.5 g) was adsorbed in silica gel (10 g normal phase), applied to a glass column with silica gel (5.0 × 40 cm, 30 g, 0.04–0.06-mm mesh, Merck, Darmstadt Germany), and eluded with hexane/ethyl acetate with 5% ascendant polarity, collecting 71 subfractions of 50 mL each. Subsequently, the subfractions were concentrated in a rotary evaporator under reduced pressure and grouped according to their similarity by TLC into 17 subfractions (C1F1-C1F17). The C1F9 fraction (834 g) was absorbed with silica gel (RP-18, Merck, Darmstadt, Germany) and fractionated on a glass column (350 × 20 mm) packed with silica gel (10 g, RP-18, Merck, Darmstadt, Germany). The elution system was water; acetonitrile, using four mixtures: 70:30; 65:35, 40:60; and methanol (100%), and obtained 16 sub-fractions of volumes of 10 mL. In fraction 3 and 4, compound (**1**) was isolated and identified by NMR and MS (see Appendix A). Compounds (**2**–**4**) and (**6**) were identified by comparison of HPLC data with standards and luteolin 7-O-rhamnoside (**5**) was identified by mass spectrometry (Appendix A). The HA extract, fractions and subfractions were analysed by TLC on silica gel 60 under light at 254 and 360 nm (Appendix A). All structures identified in the bioactive sub-fractions are shown in Figure 3.

### 4.6. HPLC-PDA and MS

The HA extract, fractions, and subfractions were subjected to chemical analysis using a Supelcosil LC-F column (4.6 × 50 mm i.d., 5-μm particle size) for chemical separation (Sigma-Aldrich, Bellefonte, PA, USA). The mobile phase consisted of 0.5% trifluoroacetic acid aqueous solution (solvent A) and acetonitrile (solvent B). The gradient system used was as follows: 0–1 min, 0% B; 2–3 min, 5% B; 4–20 min, 30% B; 21–23 min, 50% B; 24–25 min, 80% B; 26–27 min 100% B; 28–30 min, 0% B. The flow rate was maintained at 0.9 mL/min, and the sample injection volume was 10 μL. Absorbance was measured at 330 nm. Mass spectrometry (MS) analysis was on a Waters Xevo triple quadrupole (TQD) system equipped with electrospray ionization (ESI) source (Waters, Milford, MA, USA).

### 4.7. Haemonchus contortus Eggs

The nematode eggs were obtained from faecal samples (4–50 g) of two donor lambs (20 ± 6 kg bodyweight, BW) previously subjected to monospecific infection with 350 infective larvae (L3) of *H. contortus* per kg of BW (INIFAP, strain). The sheep were housed indoors on a metabolic floor with access to a feeder and drinking trough. The animals were housed following the care/welfare guidelines of the Mexican Official Rule NOM-051-ZOO-1995. Egg recovery was performed according to the technique described by Coles et al. [31].

### 4.8. Egg Hatching Inhibition (EHI) Test

The assays were performed in 96-microtitration plates. Four repetitions and three replicates in each treatment were considered (*n* = 12). The treatments were assigned as follows: step (1) hydroalcoholic extract (HA-E), aqueous fraction (Aq-F) at 0.31−5.00 mg/mL and organic fraction (EtOAc-F) at 0.03–5.00 mg/mL; step (2) subfractions (C1F1-C1F17) at 0.06–1.00 mg/mL. Here, only the bioactive subfractions were reported (see results). For each step, two suitable negative controls (methanol 2% and distilled water) were included, and additionally, a group treated with thiabendazole (0.10 mg/mL, SIGMA^®^, St. Louis, MO, USA) was considered as a positive control. In each well, 50 µL of an aqueous suspension containing 100 ± 15 *H. contortus* eggs was deposited. Then, 50 µL aliquots of the treatments and controls were deposited, giving a total volume of 100 µL. After this process, plates were incubated at room temperature (25–28 °C) for 48 h, and hatching was stopped using Lugol’s solution. The total eggs and larvae of the first and second stage of each well were counted, and the egg hatching inhibition percentage (EHI%) for each treatment was estimated using the following formula:$$\mathrm{EHI}\%\;=\frac{\mathrm{number}\;\mathrm{of}\;\mathrm{eggs}}{\mathrm{number}\;\mathrm{of}\;\mathrm{larvae}\;+\;\mathrm{number}\;\mathrm{of}\;\mathrm{eggs}}\times 100$$

### 4.9. Statistical Analysis

The EHI percentage of treatments was normalised using a square root transformation and analysed based on a completely random design using ANOVA through a general linear model in SAS. Means were compared among treatments using a Tukey test at 0.05 significance. The treatments with a concentration-dependent effect were subjected to regression analysis to estimate the effective concentrations 50 and 90 (EC_50_ and EC_90_), using the PROBIT procedure [32].

## 5. Conclusions

The results of this study showed that the *P. dulce* leaves possess ovicidal effects against *H. contortus.* The mixture of ferulic acid, coumaric acid, luteolin 7-O-rhamnoside, and quercetin, present in the subfractions C1F9, C1F11, and C1F13, could be responsible for the anthelmintic effect. However, additional studies should be performed in order to elucidate/confirm the nematocidal effect of these compounds. The use of this plant species could be an option for the control of gastrointestinal nematodes in ruminants against the background of environmental sustainability. Nevertheless, in vivo studies with sheep experimentally and naturally infected with *H. contortus* or other GIN will be necessary to confirm these findings.

## Figures and Tables

**Figure 1 plants-11-02555-f001:**
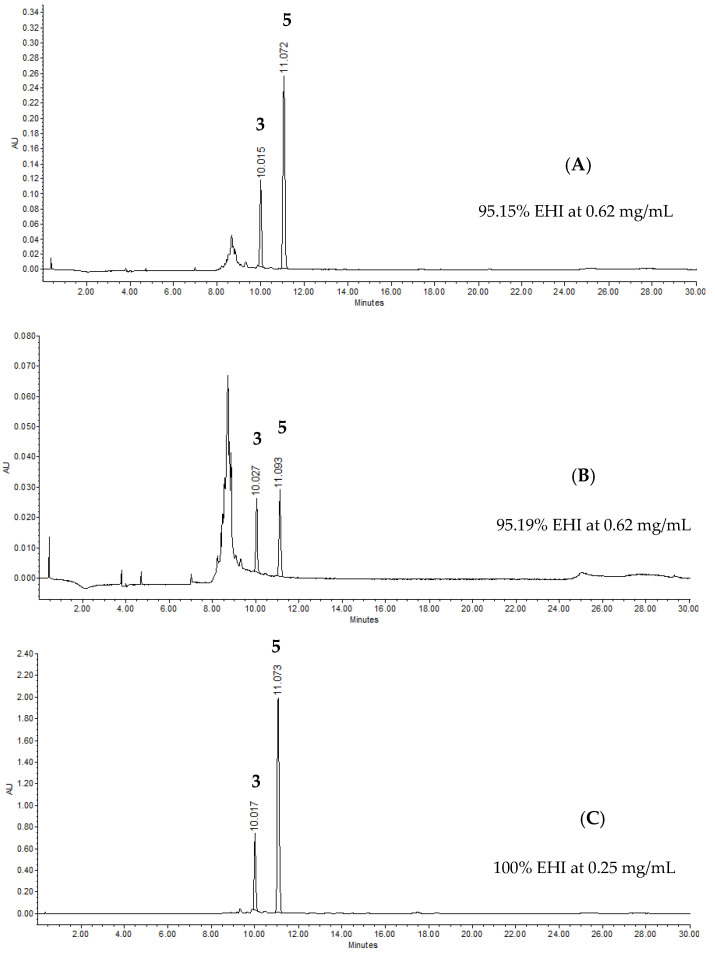
HPLC analysis of *Pithecellobium dulce* leaves showing egg hatching inhibition of *Haemonchus contortus* eggs. (**A**) hydroalcoholic extract (HA-E), (**B**) aqueous fraction (Aq-F) and (**C**) organic fraction (EtOAc-F). Coumaric acid (**3**) and luteolin 7-O-rhamnoside (**5**).

**Figure 2 plants-11-02555-f002:**
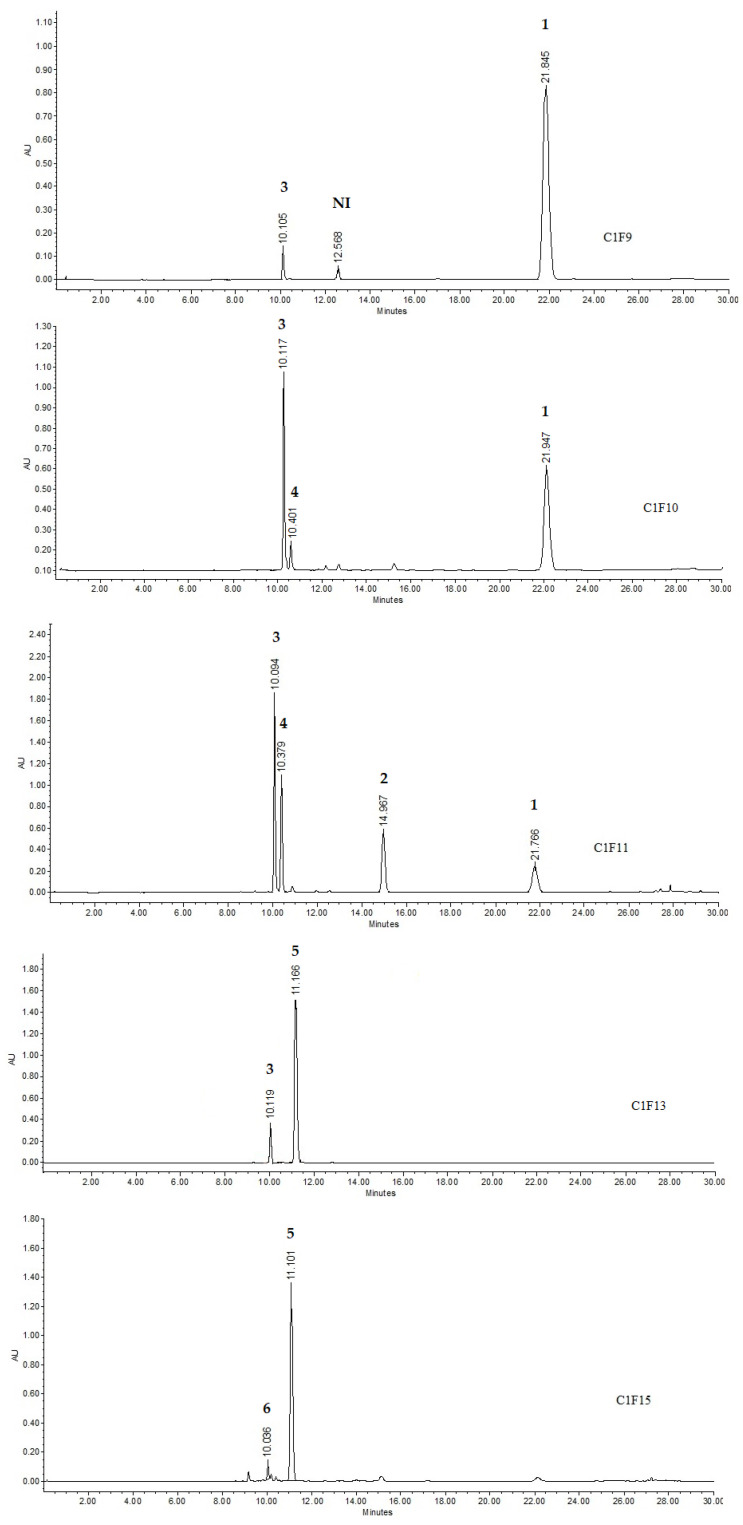
HPLC analysis of subfractions obtained from EtOAc-F, indicating the presence of kaempferol (**1**), quercetin (**2**), coumaric acid (**3**), ferulic acid (**4**), luteolin 7-O-rhamnoside (**5**), quercetin 3-O-rhamnoside (**6**) and caffeoyl derivate (NI). NI not identified.

**Figure 3 plants-11-02555-f003:**
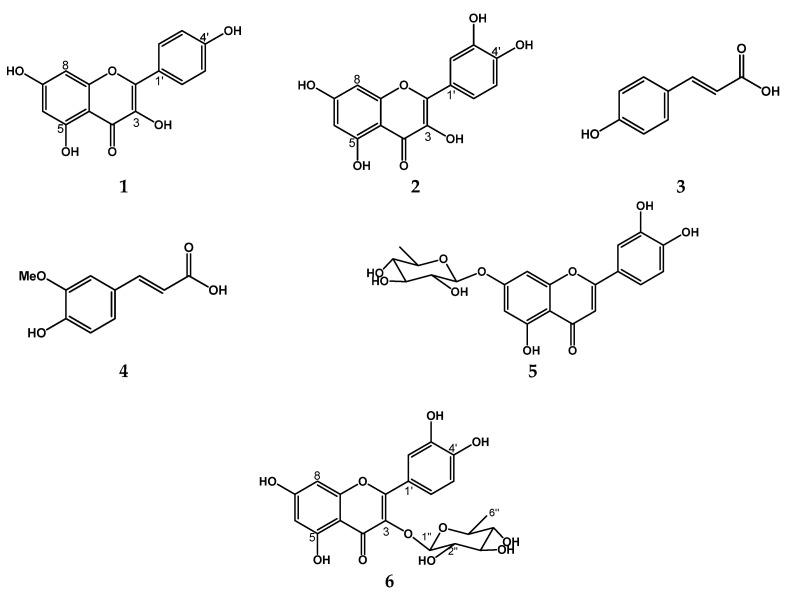
Structure identified in the bioactive subfractions from *Pithecellobium dulce* leaves.

**Table 1 plants-11-02555-t001:** *Haemonchus contortus* egg hatching inhibition percentage (EHI%) after 48 h exposure to a hydroalcoholic extract and two fractions from *Pithecellobium dulce* leaves.

Treatments	Means of Eggs and Larvae (L_1_ or L_2_) Recovered		EHI% ± s.d.
	Eggs	Larvae	
Distilled water	7.12	119.56	3.94 ± 6.61 ^e^
Methanol (2%)	7.25	111.68	3.99 ± 2.97 ^e^
Thiabendazole (0.1 mg/mL)	105.37	0	100 ^a^
Hydroalcoholic extract (HA-E, mg/mL)			
5.0	99.09	2	98.57 ± 3.52 ^a^
2.5	73.5	15.75	97.52 ± 5.92 ^ab^
1.25	54.87	36.37	96.21 ± 4.83 ^ab^
0.62	114.62	5.5	95.15 ± 5.56 ^ab^
0.31	109.12	75.75	58.63 ± 10.09 ^c^
Aqueous fraction (Aq-F, mg/mL)			
5.0	125.28	3.14	96.87 ± 4.04 ^ab^
2.5	138.36	22.18	94.42 ± 7.71 ^ab^
1.25	151.75	1.25	98.47 ± 1.87 ^ab^
0.62	125.25	9.37	92.19 ± 7.57 ^ab^
0.31	59.37	60.12	53.53 ± 13.24 ^c^
Organic fraction (EtOAc-F, mg/mL)			
5.0	120.08	0	100 ^a^
2.5	107.00	0	100 ^a^
1.25	124.50	0	100 ^a^
0.62	154.62	1	99.94 ± 0.16 ^a^
0.31	129.00	0	100 ^a^
0.15	121.25	22.37	88.24 ± 13.76 ^b^
0.07	61.25	173.75	25.87 ± 6.06 ^d^
0.03	41.51	173.75	18.95 ± 2.80 ^d^

^abcde^ Means with different literal in the same column indicate statistical differences (*p* < 0.05), s.d = standard deviation.

**Table 2 plants-11-02555-t002:** Effective concentrations required to inhibit 50% and 90% of *Haemonchus contortus* egg hatching after 48 h exposure to a hydroalcoholic extract and two fractions from *Pithecellobium dulce* leaves.

Treatments	EC_50_ (mg/mL)	Confidence Interval (95%)		EC_90_ mg/mL	Confidence Interval (95%)	
		Lower	Upper		Lower	Upper
HA-E	0.15	0.12	0.18	3.34	2.84	4.04
Aq-F	0.15	0.12	0.18	1.63	1.5	1.78
EtOAc-F	0.08	0.083	0.089	0.14	0.13	0.14

EC = effective concentration, HA-E = hydroalcoholic extract, Aq-F = aqueous fraction, EtOAc-F = organic fraction.

**Table 3 plants-11-02555-t003:** *Haemonchus contortus* egg hatching inhibition percentage (EHI%) after 48 h exposure to subfractions obtained from the EtOAc fraction from *Pithecellobium dulce* leaves.

Subfractions/Controls	Concentration (mg/mL)	Means of Eggs and Larvae (L1 or L2) Recovered		EHI% ± s.d.
		Eggs	Larvae	
C1F6	1.00	113.25	0	100 ^a^
	0.50	111.50	0	100 ^a^
	0.25	97.00	20.25	83.33 ± 6.68 ^cd^
	0.12	34.75	71.75	33.03 ± 6.87 ^e^
	0.06	105.00	96.50	9.67 ± 5.11 ^hij^
C1F9	1.00	94.50	0	100 ^a^
	0.50	82.00	0	100 ^a^
	0.25	93.37	0	100 ^a^
	0.12	119.5	0.75	99.50 ± 0.57 ^a^
	0.06	115.75	16.25	88.00 ± 2.58 ^bc^
C1F10	1.00	44.50	0	100 ^a^
	0.50	47.75	0	100 ^a^
	0.25	70.87	0	100 ^a^
	0.12	79.50	17.75	82.20 ± 3.32 ^cd^
	0.06	15.25	98.75	13.38 ± 0.68 ^ghi^
C1F11	1.00	85.75	0	100 ^a^
	0.50	76.75	0	100 ^a^
	0.25	88.62	0	100 ^a^
	0.12	121.25	1.50	98.84 ± 0.90 ^a^
	0.06	100.25	28.50	78.15 ± 2.47 ^cd^
C1F13	1.00	128.25	0	100 ^a^
	0.50	95.25	0	100 ^a^
	0.25	87.67	0	100 ^a^
	0.12	116.00	0	100 ^a^
	0.06	96.05	26.00	78.42 ± 6.42 ^cd^
C1F15	1.00	54.75	9.00	85.90 ± 6.48 ^c^
	0.50	13.00	54.50	20.00 ± 5.89 ^fg^
	0.25	13.87	70.75	15.64 ± 9.06 ^fgh^
	0.12	6.00	69.75	7.91 ± 8.43 ^hij^
	0.06	0.00	100.75	0.00 ^j^
C1F18 (kaempferol)	1.0	1.25	102.25	1.27 ± 1.13 ^j^
Methanol (2%)	2%	6.75	67.75	8.26 ± 8.50 ^hij^
Thiabendazole (0.1 mg/mL)	0.5%	12.37	100	100 ^a^

^abcdefhij^ Means with different literal in the same column indicate statistical differences (*p* < 0.05), s.d. = standard deviation.

**Table 4 plants-11-02555-t004:** Effective concentrations required to inhibit 50% and 90% of *Haemonchus contortus* egg hatching after 48 h exposure to subfractions obtained of the EtOAc fraction.

Treatments	EC_50_ mg/mL	Confidence Interval (95%)		EC_90_ mg/mL	Confidence Interval (95%)	
		Lower	Upper		Lower	Upper
C1F6	0.16	0.15	0.17	0.29	0.28	0.31
C1F9	0.03	0.02	0.04	0.06	0.06	0.07
C1F10	0.09	0.08	0.09	0.13	0.12	0.14
C1F11	0.04	0.03	0.04	0.07	0.07	0.08
C1F13	0.03	0.03	0.03	0.08	0.07	0.08
C1F15	0.67	0.62	0.72	1.30	1.14	1.62

EC = effective concentration.

## Data Availability

Data are contained within the article.

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
