# Peer review of "Phenolic Acids and Flavonoids from Pithecellobium dulce (Robx.) Benth Leaves Exhibit Ovicidal Activity against Haemonchus contortus"

_plants, 2022, doi:10.3390/plants11192555_

Round 1

Reviewer 1 Report

Manuscript entitled “Phenolic acids and flavonoids from Pithecellobium dulce (Robx.) Benth leaves exhibit ovicidal activity against Haemonchus contortus”

The objective of this study was to evaluate the ovicidal activity of a hydroalcoholic extract (HA-E) and its fractions (aqueous, Aq-F and organic EtOAc-F) from P. dulce leaves, as well as, subfractions (C1F1–C17) obtained from EtOAc-F against Haemonchus contortus eggs. The results indicated that P. dulce leaves exhibit a potent anthelmintic activity and contain bioactive compounds able to inhibit egg hatching in H. contortus. Therefore, this plant could be used for the control of gastrointestinal nematodes in small ruminants.

There are some problems, which must be solved before it is considered after major revision for publication.

1.       The chromatographic conditions of each component detected by HPLC, such as detection wavelength, mobile phase, flow rate, column temperature, etc. we all know that the components will be showed different absorption in different wavelengths.

2.       In 4. Materials and Methods part, 4.6. HPLC-PDA and CG-MS analysis, what is CG-MS?

3.       In 4. Materials and Methods part, the chromatograms of the reference substance should be provided, the standards number should be added for Kaempferol, quercetin, coumaric acid, ferulic acid and quercetin 3-O-rhamnoside, etc.

4.       The author mentioned the TLC, but I cannot find the result of TLC, for example, the Rf value. The author also mentioned CG-MS analysis, I wondered what CG-MS is, and I cannot find the relevant results about it.

5.       In Conclusions part, the authors explained “The results of this study show that the P. dulce leaves possess ovicidal effects against H. contortus..”  In my opinion, it is hasty to say “The mixture of ferulic acid, coumaric acid, luteolin 7-O-rhamnoside, and quercetin, present in the subfractions C1F9, C1F11, and C1F13, was responsible for the anthelmintic effect”, it is better if the author can supplement some activity experiments for every standards of ferulic acid, coumaric acid, luteolin 7-O-rhamnoside, and quercetin to verify whether the compounds or fractions works well.

6.       Fig 1 and Fig 2 are not very clear; I suggested the author should redraw them again with origin software. Especially in Fig 2, there is a Y axis in C1F10 and it is not very aesthetic.

7.       In Table 4, it is unreasonable that EC90 Confidence interval of C1F6 the lower is 0.78 and the upper is 0.31.

8.       In Line 109, 110, what’s the CE50?

9.       The quality of English needs improving. It is noted that your manuscript needs careful editing by someone with expertise in technical English editing paying particular attention to English grammar, spelling, and sentence structure so that the goals and results of the study are clear to the reader. For example, in Line 356, “show” should be “showed”, the authors should check the manuscript carefully and made the right tense.

10.   Please check carefully the format of references.

Author Response

Reviewer 1

The objective of this study was to evaluate the ovicidal activity of a hydroalcoholic extract (HA-E) and its fractions (aqueous, Aq-F and organic EtOAc-F) from P. dulce leaves, as well as, subfractions (C1F1–C17) obtained from EtOAc-F against Haemonchus contortus eggs. The results indicated that P. dulce leaves exhibit a potent anthelmintic activity and contain bioactive compounds able to inhibit egg hatching in H. contortus. Therefore, this plant could be used for the control of gastrointestinal nematodes in small ruminants. There are some problems, which must be solved before it is considered after major revision for publication.

Observations and/or recommendations

Response

1. The chromatographic conditions of each component detected by HPLC, such as detection wavelength, mobile phase, flow rate, column temperature, etc. we all know that the components will be showed different absorption in different wavelengths.

After analysing your comment, we noticed that UV spectra were not described in Results section. Therefore, we have included this missing information in the new version of our manuscript.

Regarding the information about HPLC, such as wavelength, mobile phase, columns, systems, etc. this information is now included in Materials and Methods sections (4.6-subsection) (highlighted in yellow). 

2. In 4. Materials and Methods part, 4.6. HPLC-PDA and CG-MS analysis, what is CG-MS?

Thanks for catching the typo. It should be GC-MS.

This correction was done.

3. In 4. Materials and Methods part, the chromatograms of the reference substance should be provided, the standards number should be added for Kaempferol, quercetin, coumaric acid, ferulic acid and quercetin 3-O-rhamnoside, etc.

The chromatograms and their UV spectra of standard references were included as supplementary material.

4. The author mentioned the TLC, but I cannot find the result of TLC, for example, the Rf value. The author also mentioned CG-MS analysis, I wondered what CG-MS is, and I cannot find the relevant results about it.

TLC missing plates have been inserted in Figures 1 and 2 in the manuscript. 

The correct spectrometric analysis was considered in the manuscript.

5. In Conclusions part, the authors explained “The results of this study show that the P. dulce leaves possess ovicidal effects against H. contortus..” In my opinion, it is hasty to say “The mixture of ferulic acid, coumaric acid, luteolin 7-O-rhamnoside, and quercetin, present in the subfractions C1F9, C1F11, and C1F13, was responsible for the anthelmintic effect”, it is better if the author can supplement some activity experiments for every standards of ferulic acid, coumaric acid, luteolin 7-O-rhamnoside, and quercetin to verify whether the compounds or fractions works well.

Authors are very grateful for these recommendations.

Our conclusion was re-structured according to your comment and we propose that future studies should have to be performed in order to dilucidated the nematocidal effect of these compounds.

6. Fig 1 and Fig 2 are not very clear; I suggested the author should redraw them again with origin software. Especially in Fig 2, there is a Y axis in C1F10 and it is not very aesthetic.

Authors agree with this suggestion. In the new manuscript figures 1 and 2 were improved.

7. In Table 4, it is unreasonable that EC90 Confidence interval of C1F6 the lower is 0.78 and the upper is 0.31.

We had typo in this confidence interval. This mistake was corrected in the new version.

The lower value “0.78” was replaced by “0.28”.

8. In Line 109, 110, what’s the CE50?

The typo: CE50 was replaced by “EC50”

9. The quality of English needs improving. It is noted that your manuscript needs careful editing by someone with expertise in technical English editing paying particular attention to English grammar, spelling, and sentence structure so that the goals and results of the study are clear to the reader. For example, in Line 356, “show” should be “showed”, the authors should check the manuscript carefully and made the right tense.

The new version was checked by an English native speaker

10. Please check carefully the format of references.

This observation was attended

Reviewer 2 Report

The authors developed a study of separation and identification of phenolic acids and flavonoids from Pithecellobium dulve (Robx.) Benthe leaves and evaluated their ovicidal activity against Haemonchus contortus. For this, they employed several biological and chemical techniques. The study is interesting and has scientific relevance, but some improvements must be made. 

1) Improve the introduction.

2) Some methodology must be described and discuss in detail. For example: The HA extract, fraction and subfractions were used for the separation and identification of compounds, but the chemical discussion is poor. Besides that, there are some improvements that must be made.

3) What is the purity of thiabendazole and where was it purchased?

4) Show some fragmentation mechanisms of the ions of substances.

5) In research with biological activity, it is important to know at least estimated amount of substances in extracts, fractions and subfractions. Why weren't the concentrations of the substances in these samples estimated?

6) Attach the experimental mass spectra, total ion chromatograms and uv spectra of the compounds as supplementary material. It is also important to attach the total ion chromatograms.

Author Response

Reviewer 2

The authors developed a study of separation and identification of phenolic acids and flavonoids from Pithecellobium dulve (Robx.) Benthe leaves and evaluated their ovicidal activity against Haemonchus contortus. For this, they employed several biological and chemical techniques. The study is interesting and has scientific relevance, but some improvements must be made. 

Observations and recommendations

Response

1) Improve the introduction.

The introduction was improved

2) Some methodology must be described and discuss in detail. For example: The HA extract, fraction and subfractions were used for the separation and identification of compounds, but the chemical discussion is poor. Besides that, there are some improvements that must be made.

Additional details description on this regard are now included under Materials & Methods. (Highlighted in yellow). 

3) What is the purity of Thiabendazole and where was it purchased?

The purity of Thiabendazole is ≥ 98.0 %. This anthelmintic was purchased from SIGMA®, USA.

This information was included in the new version of the manuscript -Materials & Method section.

4) Show some fragmentation mechanisms of the ions of substances.

MS Spectra was included as supplementary material and data of this experiment was added in results section.

5) In research with biological activity, it is important to know at least estimated amount of substances in extracts, fractions and subfractions. Why weren't the concentrations of the substances in these samples estimated?

Percentages of area under curve for each identified compound in extract and fractions was included in the results section.

6) Attach the experimental mass spectra, total ion chromatograms and uv spectra of the compounds as supplementary material. It is also important to attach the total ion chromatograms.

This missing information was included as supplementary material in the new manuscript.

Round 2

Reviewer 1 Report

1. It doesn’t make sense because the quality of TLC is really not good. Every compound should be marked on TLC plate.  There is no reference substance on TLC. The authors should do the TLC experiments again and add the reference substance on TLC.

2. In Line 111-112, CE50 should be changed to EC50.

3. In Line 122, Line 323, nm should be changed nm.

Author Response

Reviewer 1

Observations and recommendations

Response

1. It doesn’t make sense because the quality of TLC is really not good. Every compound should be marked on TLC plate.  There is no reference substance on TLC. The authors should do the TLC experiments again and add the reference substance on TLC.

We have carried out the TLC of the extract and fractions compared with standards. Data were included in the supplementary data (S11-S17)

2. In Line 111-112, CE50 should be changed to EC50.

Suggestion attended

3. In Line 122, Line 323, nm should be changed nm.

Suggestion attended

Reviewer 2 Report

The authors made the improvements during the first phase of the review. In this way, the manuscript can be published.

Author Response

The authors made the improvements during the first phase of the review. In this way, the manuscript can be published.

Response

Thank you for your review.